# Biogeographical Relationships and Diversity in the Peruvian Flora Reported by Hipólito Ruiz and José Pavón: Vegetation, Uses and Anthropology

**DOI:** 10.3390/biology12020294

**Published:** 2023-02-13

**Authors:** Juan Miguel Arias-Gámez, Eliana Linares-Perea, José Alfredo Vicente-Orellana, Antonio Galán-de-Mera

**Affiliations:** 1Laboratorio de Botánica, Facultad de Farmacia, Universidad San Pablo-CEU, CEU Universities, Urbanización Montepríncipe, Boadilla del Monte, 28660 Madrid, Spain; 2Fundación Estudios Fitogeográficos del Perú, Arequipa 04002, Perú

**Keywords:** bioclimatology, ethnobotany, history, Peru, vegetation

## Abstract

**Simple Summary:**

The expedition started by Hipólito Ruiz and José Pavón from Spain to the territories of Peru, Chile and Ecuador in the 18th century yielded a total of 6493 plants from Peru, of which 2327 were used in medicine, cosmetics, food or materials. Information was obtained from their published works, diaries, manuscripts and plant collection. Using a bioclimatic model, we present the types of vegetation from which these plants originate (from the coastal desert to the Amazonian forests), as well as the ethnic groups from which they might have come. Using statistical analyses, we were able to discover which vegetation types were the most explored and the areas from which the most commonly used plants originated.

**Abstract:**

The Royal Spanish Botanical Expedition to the Viceroyalty of Peru in the 18th century was one of the most important European expeditions to American territories. Using the herbarium sheets of Ruiz and Pavón (Royal Botanical Garden of Madrid) and their edited works, manuscripts and expedition diaries, we have constructed a database of the collected and observed flora, which has served as the basis for a map containing all of the Peruvian localities of the expedition. Based on the method of bioclimatic belts and our own observations, we have deduced to which type of vegetation the flora studied in the expedition belongs. The uses of the flora per locality were studied, as well as the ethnic groups involved in the different localities. By using a Principal Component Analysis, we have obtained the distribution of the bioclimatic belts whose vegetation was the most explored. In order to observe the bioclimatic tendency of plant uses, a Detrended Correspondence Analysis (DCA) was conducted to identify the distribution of localities with the highest frequencies of plant uses. The expedition’s explorations focused on the most humid areas of the thermo- and mesotropical belts, from where a large number of plants with practical uses were obtained.

## 1. Introduction

Peru is one of the countries with the greatest diversity of vascular plants in the world [1] and the third most diverse in South America after Brazil and Colombia, with just over 19,000 species [2]. Its geography is divided into three major natural regions [3]: the coast, with a very arid desert; the highlands, starting with the first foothills of the Andes, ranging from communities of succulent plants (Cactaceae) to the grasslands of the puna above 4000 m asl and up to the summits, which, in many cases, exceed 5000 m asl; and the Amazonian slope of the Andes, which begins with very humid forests that continue into the Amazon basin, from about 1000 m asl, with a diversity of up to 300 tree species per hectare [4].

The study of this diversity was initiated by the Royal Spanish Botanical Expedition to the Viceroyalty of Peru (RSBEVP) (1777–1815), which was one of the most important scientific explorations of the time. Its main objective was to collect and describe the flora of the current territories of Peru, Chile and Ecuador, but also to obtain plant uses and popular names. Sponsored by King Carlos III of Spain, it was led by Hipólito Ruiz, a botany enthusiast connected with the Royal Botanical Garden of Madrid. He was accompanied by José Antonio Pavón, a young apprentice in the royal apothecary, and the French doctor and naturalist Joseph Dombey, as well as illustrators José Brunete and Isidro Gálvez [5]. Many of their findings are still unpublished, while others were published in several volumes [6,7,8,9,10,11]. Most of the herbarium materials and drawings from the expedition are digitized in the Global Plants database on the JSTOR portal [12].

The expedition traveled through most of the Peruvian territory in different stages, from the coastal desert to certain parts of Amazonia in the center of the country, even making some forays northward (Cajamarca mountains) and southward to the Atiquipa hills, in the department of Arequipa. Hipólito Ruiz and José Pavón collected plants in central and northern Peru, around Lima, the coastal areas of Chancay and Huaura, and the regions of Canta, Tarma, Jauja, Pasco and Huánuco. Between 1788 and 1815, the expedition was joined by Juan José Tafalla and, somewhat later, by Juan Agustín Manzanilla, who provided the material from the mountains of Monzón (Huamalíes), Huánuco, Pozuzo (central Peru) and the regions of Ica and Arequipa (southern Peru).

This led the expeditioners to collect and study plants from most of the habitats of Peru, collections that were concentrated especially in the Royal Botanical Garden of Madrid, and from these, most of the diaries and works were written [13,14,15,16,17,18,19].

Some Peruvian authors also made reference to this initial expedition of Linnaean scientific botany in these territories [20,21,22] although without the desired detail, focusing on the new post-Hispanic scientific contributions. Nevertheless, the latter author summarized the expedition’s route, indicating some plants from the coast, such as *Nolana* sp. and *Sapindus saponaria* L., and others from the lowland forest, such as *Chondodendron tomentosum* Ruiz & Pav. and *Cinchona nitida* Ruiz & Pav.

From 1815 onward, when the last consignment of materials was sent to Spain, the death of Ruiz in 1816 and various factors, both internal and external to the expedition, including the events of the advent of the Peruvian Republic, marked the decline of the expedition over the years until its end in 1835. It is worth noting that Juan José Tafalla, with his republican leanings, published the first description of coca in the pro-republican newspaper “El Mercurio Peruano”, outlining the idea of the dissemination of advances in Peruvian Botany [23,24], and was the founder of the first American botanical garden and the first chair of Botany at the Universidad Nacional Mayor de San Marcos [25].

However, Northern Hemisphere authors who wrote about the expedition, perhaps due to their ignorance of Peruvian flora and vegetation, did not synthesize the environment that surrounded Ruiz and Pavón and their collaborators, nor the large amount of information derived from their materials, and neither did Peruvian authors because they did not consult the herbarium and the extensive bibliography derived from it, which are undoubtedly the origin of modern Peruvian flora.

The collections of the RSBEVP are very extensive, and with all of the information obtained, both from the materials preserved in Madrid and from the observations made in Peru, the aims of this work were to map the localities of the entire expedition in Peru (1), to determine the types of vegetation from which the plants were obtained (2), as well as the uses of these plants, linking them to the types of vegetation (3), and going even further, to identify the ethnic groups from which the plants could have come and their uses (4).

## 2. Materials and Methods

### 2.1. Building Databases

To start the study, a base of localities compiled from the herbarium of Ruiz and Pavón (MA according to Index Herbariorum abbreviations [26]) and from their edited works [6,7,8,9,10,11], diaries and unpublished material was elaborated and transposed into their current toponymies (Appendix A). Cartographic resources from the 18th century, cartographic software and Peruvian toponymies published since the 19th century were used, including the statistical dictionary by Paz Soldán [27] (Appendix A). The decimal coordinates of all localities were looked up (Appendix A). These were used to construct digitized maps of the localities mentioned in the expedition, which were generated using the software Quantum Geographic Information System (QGIS) version 3.10 A Coruña [28].

For each locality, the list of plants collected or observed during the expedition was compiled, and the nomenclature of those indicated in the text was updated according to the herbarium labels of the material studied by other authors and to the IPNI portal [29]. There are species names that were not found and others that do not geographically match their distribution, as they are from outside the territory studied. In these cases, the genus rank was used. In addition, the uses assigned to the plants by the indigenous people of the localities through which the expedition passed were noted and translated according to Cook [30]. Cook established 10 groups belonging exclusively to plants: food (including beverages, for humans only), food additives (processing agents and other additive ingredients used in food preparation), animal food (fodder and feed for vertebrate animals only), materials (wood, fibers, cork, cane, tannins, latex, resins, gums, waxes, oils, lipids, etc., and their by-products), fuels (wood, charcoal, petroleum substitutes or fuel alcohols), social uses (plants used for social purposes that cannot be defined as food or medicine, e.g., chewing and smoking materials, narcotics, hallucinogens and psychoactive drugs, contraceptives and abortifacients, and plants for ritual or religious purposes), vertebrate poisons (plants that are poisonous to vertebrates, both accidental and useful for hunting and fishing), non-vertebrate poisons (both accidental and useful poisons, e.g., molluscicides, herbicides and insecticides), medicines (both human and veterinary medicines) and environmental uses (intercropping and nurse crops, ornamental plants, barrier hedges, shade plants, windbreaks, soil improvers, revegetation and erosion control plants, sewage treatment plants, and indicators of the presence of metals, pollution or groundwater). To Cook’s uses, we add that of cosmetics/perfumery (plants used for skin or hair improvement and beautification purposes and plants used for their scent), as we know that the expeditioners knew of plants that the Incas and pre-Incas infused for these purposes [31]. For simplicity, cosmetics/perfumery is treated as “cosmetics” in the Appendix A, text and figures.

For each locality, there are a number of plants with specific uses, which are listed in Appendix A.

To determine which type of vegetation was present at each expedition locality, the Rivas-Martínez bioclimatic belt model [32] was applied, including at least 30 years of data, using the databases Meteoblue [33] and Chelsa [34]. This bioclimatic model is based on thresholds of the thermicity index (It) versus the intervals of annual precipitation (P in mm) coincident with altitudinal and latitudinal areas of flora and vegetation, called bioclimatic belts. The thermicity index is calculated on the basis of the mean annual temperature (T in C) and the average maximum (M) and minimum (m) temperatures of the coldest month (It = (T + M + m) 10). Bioclimatic belts coincide with natural plant associations, and for Peru, six bioclimatic belts were defined and mapped [35]: infratropical (It > 690), thermotropical (It 490 to 690), mesotropical (It 320 to 490), supratropical (It 160 to 320), orotropical (It 50 to 160) and cryorotropical (It < 50). P (in mm) intervals or ombrotypes are the following: ultra-hyperarid (*p* < 5), hyperarid (5 to 30), arid (31 to 100), semiarid (101 to 300), dry (301 to 500), subhumid (501 to 900), humid (901 to 1500), hyperhumid (1501 to 2500) and ultra-hyperhumid (>2500).

The bioclimatic zones and intervals of annual precipitation correspond to different types of plant communities formed by the combination of specific flora (Table 1).

### 2.2. Statistical Analysis

Principal Component Analysis (PCA) was performed to determine how the localities with useful plants from the expedition are distributed among bioclimatic belts and ombrotypes. To observe the bioclimatic trends of plant uses, a Detrended Correspondence Analysis (DCA) plot was generated, where the axes are the uses (Axis 1) and the localities where the expedition collected or observed plants with a higher frequency of different uses (Axis 2). Statistical analyses were performed with the program PAST 4.07b [37].

### 2.3. Map of Ethnic Groups of Peru

The INDEPA map [38] was used to show the territories of the ethnic groups whose plant uses were reported during the expedition, as this map locates the ancestral settlements of all of the indigenous people of Peru.

## 3. Results

### 3.1. Expedition Localities

In Figure 1A, we present all of the localities of the expedition (Appendix A), relating them to the bioclimatic belts and the ombrotypes (Figure 1B,C). The localities are collections or observation points that appear in the herbarium, works, diaries and unpublished manuscripts of the RSBEVP.

### 3.2. The Plants of the Expedition and the Vegetation of Origin

The RSBEVP reported a total of 6493 plants from 191 localities of Peruvian flora (Appendix A). Of these, the uses of 2327 were reported from 128 localities in Peru (Appendix A).

Most of the localities visited by the expedition correspond to the thermotropical belt (Figure 2A); i.e., they correspond to the coastal desert, the first foothills of the western Andes or the eastern thermal Andean forests. In addition, most of them belong to the humid and hyperhumid ombrotypes (Figure 2B).

The number of plants within the hyperhumid thermotropical belt is the highest, as can be seen in Figure 3, and in turn, this bioclimatic belt is also the one with the greatest diversity of plants observed, from the ultra−hyperarid to the ultra-hyperhumid ombrotypes. In the thermotropical zone, the highest frequency of plants is found in the hyperhumid and hyperarid ombrotypes. For example, the expeditioners collected *Erythroxylum coca* Lam. in a hyperhumid ombrotype (San Juan de Cocheros, Huánuco, Peru), and the vegetation type of this species is rainforest (Table 1). On the other hand, in the hyperarid thermotropical belt, they collected *Loasa bipinnatifida* Ruiz & Pav. (= *Nasa urens* (Jacq.) Weigend), *Loasa nitida* Desr. (Lomas de Lachay, Lima, Peru), *Nolana prostrata* Ruiz & Pav. (= *Nolana humifusa* (Gouan) I.M.Johnst.) and *Narcissus amancaes* Ruiz & Pav. (= *Ismene amancaes* (Ruiz & Pav.) Herb.) (Lima, Peru), with the latter having social and medical uses. These species are part of the ephemeral vegetation of the coastal desert (Table 1).

To a lesser extent, Ruiz and Pavón also made a considerable number of collections and observations in the dry to hyperhumid mesotropical belt (Figure 3). For example, in the subhumid ombrotype, they collected *Polypodium calaguala* Ruiz (= *Campyloneurum angustifolium* (Sw.) Fée (Ayacucho, Peru), commonly used in medicine, providing evidence of mesophytic forests (Table 1), while in the dry ombrotype, they reported *Buddleja incana* Ruiz & Pav., used for construction materials and utensils, *Cervantesia tomentosa* Ruiz & Pav. (= *Cervantesia bicolor* Cav.) with the same uses in addition to its use as food, and *Jarava ichu* Ruiz & Pav., used for animal food, materials and fuels (Huarochirí, Lima, Peru), indicating mesophytic forests and grasslands. The hyperhumid mesotropical belt is of great interest because it contains rainforest plants such as *Cinchona* species (*C. glabra* Ruiz, *C. hirsuta* Ruiz & Pav.), which are used in medicine and for obtaining materials (San Pablo de Pillao, Huánuco, Peru) (Appendix A). Other interesting species in these forests are *Alstroemeria tomentosa* Ruiz & Pav. (= *Bomarea ovata* (Cav.) Mirb.), used for food, *Canna iridiflora* Ruiz & Pav., used for ornamental and food uses, and *Arbutus maccha* Ruiz & Pav. (= *Gaultheria myrsinoides* Kunth), used as food and vertebrate poisons.

Ruiz and Pavón also collected plants from the supratropical belt, which is especially humid (Figure 3), where we also find eastern Andean rainforest plants such as *Cinchona viridiflora* Pav. (= *Pimentelia glomerata* Wedd.) (Piñayoj, Huánuco, Peru), *Acrostichum huacsaro* Ruiz (= *Elaphoglossum huacsaro* (Ruiz) Christ), used in medicine, *Berberis glauca* DC. and *Bocconia frutescens* L., used for making utensils and dyeing cloth (Chaglla, Huánuco, Peru).

The orotropical and infratropical belts were barely represented in the expedition (Figure 3). In the orotropical belt, only 39 plants were found in the humid ombrotype, and only 1 was in the subhumid ombrotype. All of them belonged to the grassland biome. In the humid ombrotype, we can cite *Rauvolfia flexuosa* Ruiz & Pav. (= *Citharexylum spinosum* L.) (Quivilla, Huánuco, Peru), *Swertia umbellata* Ruiz & Pav. (= *Halenia umbellata* (Ruiz & Pav.) Gilg), *Cactus* Ruiz & Pav. (= probably *Austrocylindropuntia floccosa* (Salm-Dyck) F. Ritter) and *Valeriana coarctata* Ruiz & Pav., used in medicine (Meseta de Bombón, Junín-Pasco, Peru). The only subhumid orotropical species is *Cactus melocactus* L. (= *Oroya peruviana* (K. Schum.) Britton & Rose, and also probably *Matucana haynii* (Otto ex Salm-Dyck) Britton & Rose) (Yauli, Junín, Peru) (Appendix A, Figure 4). The most represented version of the infratropical belt is the ultra−hyperhumid one, where we find a large number of plants belonging to rainforests, among which are *Cinchona micrantha* Ruiz & Pav, used medicinally, *Laurus acutifolia* Ruiz & Pav. (= *Nectandra acutifolia* (Ruiz & Pav.) Mez), used in the manufacture of materials and utensils, *Theobroma alba* Ruiz & Pav. (= *Theobroma subincanum* Mart.) and *Phytelephas macrocarpa* Ruiz & Pav, whose fruits are used for food, and other parts of the palm are used for the manufacture of materials (Pueblo Nuevo, Huánuco, Peru).

The cryorotropical belt is not represented in the expedition.

### 3.3. Distribution of Useful Plants from the Expedition

The PCA in Figure 5 shows how the localities of plants with uses are distributed among the bioclimatic belts and ombrotypes in five groups: A—thermotropical localities from ultra-hyperarid to semiarid zones of the coastal desert; B—infra- and thermotropical Andean localities (Rio Marañón in Jaén; Huánuco, near Jauja, Peru); C—meso- and supratropical dry−subhumid Andean localities (Canta in Lima, Huariaca in Pasco, Higueras and Santo Domingo de Rondós in Huánuco, Peru); D—humid supra- and orotropical localities (Tambo Sarria and Lago Lauricocha in Huánuco, Tarma in Junín, Peru); and E—humid to ultra−hyperhumid Andean montane forests and Amazonian thermo- and infratropical rainforests (Pueblo Nuevo, Tulumayo and Casapillo in Huánuco, Vitoc in Junín, Peru).

Component 1 (It) is ordered from lowest to highest on the ordinate axis, with the most negative parts of the axis corresponding to the supra− and orotropical bioclimatic belts, while the most positive parts correspond to the infra− and thermotropical zones. The negative values of Component 2 (P) represent the most arid and dry localities, while in the positive part of the axis, the most humid localities are ordered.

The DCA in Figure 6 shows the distribution of plants with uses by their localities. Localities are ordered according to the number of plants with uses in a locality. For example, Canchán (12) (Huánuco, Peru), which is subhumid mesotropical, is very close to “medicines” because it has data for three plants used in medicine but also has one plant used for “materials”, “fuels”, “vertebrate poisons”, and “cosmetics” (see Appendix A). Chiuchin (29) (Pasco, Peru), which is dry mesotropical, has up to 14 plants with uses, 3 as “materials”, 2 as “food”, 8 as “medicines” and 1 as “cosmetics”.

However, in other cases, such as Pampamarca (81) (Huánuco, Peru), which is hyperhumid thermotropical, we have only found three plants with uses, such as “materials”, “social uses” and “medicines”. Cajamarquilla (10) and La Quinua (62) (Pasco, Peru), both humid supratropical, only present one plant with uses, “fuels” and “materials”, in this case with a clear reference to the ancestral and current uses of *Polylepis racemosa* Ruiz & Pav. In contrast, Pillao (88) (Huánuco, Peru) presents many plants with different uses: 9 as “food”, 3 as “food additives”, 1 as “animal food”, 29 as “materials”, 2 as “fuels”, 1 as “vertebrate poisons”, 15 as “medicines”, 5 as “environmental uses” and 6 as “cosmetics”; this abundance corresponds to humid eastern forests (Table 1). The DCA point cloud becomes more condensed with localities where plants have more uses, which also relate to the thermo- and mesotropical belts (Figure 3) with a wide range of ombrotypes.

### 3.4. Ethnic Groups, Plants and Uses

The map in Figure 7 shows the ethnic groups of Peru and their coincidence with the localities of the expedition, and it is clearly evident that the expedition had contact with the inhabitants and learned about the uses of the plants. Along the western slope of the territory, we have the Quechua stock, but between Lima and Junín, we can observe different ethnic groups: in Lima Yauyos and Yacaru, and in Junín, Huancas, Tarumas and Xauxas. In the eastern jungles, the expedition’s localities coincide with the territories of the Pano, Arawaw and Jíbaro families, particularly with the Yanesha, Uni and Cashibo−Cacataibo ethnic groups. Most of the observations and collections within the genus *Cinchona* come from the territories of these peoples, as well as *Phytelephas macrocarpa,* which the expedition undoubtedly observed in Yanesha, Ashéninka and Uni territories.

## 4. Discussion

The herbarium materials of the expedition, edited books on the flora and unpublished manuscripts and diaries offer a great complexity of information, mainly derived from the vicissitudes of the authors, as well as from the difficult orography of Peru.

### 4.1. Plants, Bioclimatic Belts and Landscapes

In the hyperarid thermotropical belt of the Peruvian desert coast [35], the genus *Nolana* [39] is of biogeographical importance. Ruiz and Pavón cited *N. prostrata* L. (= *N. humifusa* (Gouan) I.M.Johnst. subsp. *humifusa*) in Chancay, city of Lima, Hacienda de Torreblanca (Lima), named *N. ventricosa* nom. nud, and described *N. spathulata* Ruiz & Pav. from the hills of Pongo (Arequipa) and *N. coronata* Ruiz & Pav. (= *N. humifusa* (Gouan) I.M.Johnst. subsp. *humifusa*) from the hills of Atiquipa and Camaná (Arequipa) (Appendix A). According to the unpublished manuscripts, it was Tafalla who collected in the hills of the department of Arequipa, contributing to the first knowledge of the species of the genus *Nolana*, which served as a basis for later studies [40]. Another emblematic species of the ephemeral plant communities of the hills is *Narcissus amancaes* Ruiz & Pav. (= *Ismene amancaes* (Ker Gawl.) Herb.), described in the hills of Amancaes, Chancay, Lima, Lurín and Surco (Lima). At present, this endemism does not exist in the localities where Ruiz and Pavón found it [41] due to grazing and urban expansion.

Ruiz and Pavón cited numerous species in the hyperarid thermotropical coastal belt, e.g., in Amancaes (37 species), Cercado de Lima (31), Chancay (434) and Lima (472) (Appendix A, Figure 5), where many of them were cultivated. Examples include *Ageratum conyzoides* L., which comes from Mexico, *Croton balsamifer* Jacq. (= *Croton flavens* L.) from Mesoamerica, the Caribbean and Venezuela, *Pinus pinea* L. from Europe and *Theobroma cacao* L. from Amazonia [29]. This coincides with the large list of plants compiled by González Laguna [42], which were brought to Lima from other parts of the world, including areas within Peru. From the Iberian Peninsula, Ruiz and Pavón allude to *Fumaria officinalis* L., *Geranium moschatum* (L.) L. (= *Erodium moschatum* (L.) L’Hér.) and *Myrtus communis* L., with the latter two also being listed by González Laguna [42]. However, others discussed by this author, such as *Matricaria chamomilla* L., do not appear in Ruiz and Pavón’s work, although we could include them among the plants used by the Inca culture [43].

Another important thermo−mesotropical species is *Erythroxylum coca* Lam., which Ruiz and Pavón discovered in the vicinity of Huánuco, Acomayo, Chinchao, Cuchero, Pozuzo (Tilingo) (Huánuco, Peru), and is cultivated in humid to ultra-hyperhumid areas. They also distinguished the species *Erythroxylum acuminatum* Ruiz & Pav., which extends to the inland of the infratropical Amazonia of Brazil [29] and which was recently used to clarify new species in Colombia and Panama [44]. However, *Erythroxylum coca* var. *ipadu* Plowman (Amazonian coca), which was already known to Europeans in the mid-18th century [45], is not included in the works by Ruiz and Pavón, nor is *E. novogranatense* var. *truxillense* (Rusby) Plowman, which comes from the valleys of northern Peru and was cultivated on the coast before the Incas [46].

The flora of the meso− and supratropical levels is very well represented in the expedition flora by the shrubs of *Mutisia acuminata* Ruiz & Pav. and *Periphragnos uniflorus* Ruiz & Pav. (= *Cantua buxifolia* Lam.). *Mutisia acuminata* was reported in Cajatambo, Chiuchín, Huarochirí (Lima) and Tarma (Junín), and *Cantua buxifolia* was reported in Canta and Huarochirí (Lima), Chaglla (Huánuco), Cullhuay (Lima) and Tarma (Junín) (Appendix A), which can be confirmed today [47]. In the supratropical belt of central Peru, *Polylepis racemosa* forests are also widespread [48]. During the expedition, they were found in the mountains of Panatahuas (Huánuco), Huariaca (Pasco), Cajamarquilla (Pasco), La Quínua (Pasco), San Pablo de Pillao (Huánuco) and Tarma (Junín), together with other undescribed species of the genus. At least the populations of La Quinua and Cajamarquilla (locotypic indication) still exist today (observation by E.L. and A.G.M.), below the orotropical vegetation of *Calamagrostis rigida* (Kunth) Trin. ex Steud. and *Festuca dolichophylla* J.Presl.

Perhaps the genus most studied by the expeditioners within the orotropical belt is *Valeriana*, whose species are part of the grasslands or rocky communities at high altitudes [49], such as *V. connata* Ruiz & Pav, *V. interrupta* Ruiz & Pav. and *V. pilosa* Ruiz & Pav. at Hacienda El Diezmo (Huayllay, Pasco); *V. lanceolata* Ruiz & Pav. (= *V. coarctata* Ruiz & Pav.) and *V. rigida* Ruiz & Pav. in Meseta de Bombón (Cochamarca, Pasco); *V. oblongifolia* Ruiz & Pav. in Cerro de Pasco; and *V. coarctata* Ruiz & Pav. and *V. thyrsiflora* (= *V. coarctata* Ruiz & Pav.) in Huasahuasi (Junín). However, some orotropical species of Cactaceae, cited by Ruiz and Pavón as *Cactus melocactus* L. from Yauli and la Oroya (Junín), which are *Oroya peruviana* and apparently also *Matucana haynii* (Otto ex Salm-Dyck) Britton & Rose (Appendix A, Figure 4), also attract attention.

The humid forests of eastern Peruvian, between the infra- and supratropical belts, are characterized by the relatively frequent presence of species of the genus *Cinchona* [37], whose importance extends from the expedition to the present day [50], being the subject of some publications by Ruiz [51] and Ruiz and Pavón [52] as well. Its study is one of the main achievements of their expeditions [53,54]. Among the species, *C. micrantha* is quite widespread in the Amazonian montane forests of central and southern Peru (Figure 8) in the thermo− and infratropical hyperhumid and ultra−hyperhumid bioclimatic zones with tree ferns (*Cyathea*) and palms (*Iriartea deltoidea* Ruiz & Pav.) [35]. *C. pubescens* Vahl ascends in central and northern Peru to the hyperhumid mesotropical belt (Appendix A), also accompanied by tree ferns [37].

Tafalla and Manzanilla also collected *Theobroma cacao* L. and *T. alba* Ruiz & Pav. (= *T. subincanum* Mart.) and the palm *Phytelephas macrocarpa* in the Amazonian rainforests of central Peru, recording its use as food and materials. Unknowingly, they delved into the original area of the cocoa crop [55], in whose forests we find wild cocoa today [36]. *Phytelephas macrocarpa* is widespread, with uses such as those described in the expedition diaries, as the fruit is now used as food and for roofing [56].

### 4.2. The First Ethnobotanical Expedition, Anthropology and Vegetation

One of the most interesting aspects of the expedition is that Ruiz, Pavón and Tafalla obtained ethnobotanical information on all of the vegetation types of Peru with their altitudinal distributions (Table 1). According to Schultes [57], Hipólito Ruiz became the great ethnobotanist of his time, since one of the functions assigned to him in Spain was to obtain the popular names and uses of the plants [13,18].

Although all of the unpublished and edited notes from the expedition recorded 2327 plants (Appendix A), about 5000 plants with uses are known today in Peru [58]. Few works have been written on the relationship between uses, anthropological aspects, bioactive compounds and the ecological (geographical) environments of the plants [59]. However, although the expedition initiated by Ruiz and Pavón ended two centuries ago, many of the habitats of that time still exist. It seems that Hipólito Ruiz’s warning about the deterioration and preservation of the quina forests was heeded [53]. Galán de Mera et al. [59] have observed that in northern Peru, there is a relationship between the distribution of bioactive compounds of plants and bioclimatic belts, in such a way that alkaloids are typical of the lower bioclimatic belts, phenolic compounds, flavonoids and steroids are abundant in the meso− and supratropical belts, and iridoids reach the orotropical belt. The same observation can be made for plants from the expedition related to vegetation types (Table 2).

It is clear that the members of the expedition must have had contact with certain ethnic groups in Peru, especially in the eastern jungles (Figure 7). In the diaries published by Barreiro [14] and Jaramillo [16], indigenous communities were mentioned on numerous occasions, and the localities coincide with the mapped ancestral ethnic groups [38]. Table 3 shows the correspondence between some texts of the diaries, with the names applied to the indigenous people, the locality and the mapped ethnic group. The coincidences between the expedition’s localities and the territories of the Pano, Arawaw and Jíbaro ethnic families are evident. For example, the collected specimens of *Cinchona* come from forests inhabited by these people. In the region of Pueblo Nuevo (Huánuco), in the surroundings of present-day Tingo María, we find one of the few places where the expedition reached the lowland Amazonian rainforest. Here live the Yaneshas, who still use the leaves of *Phytelephas macrocarpa,* just as they did in the time of Ruiz and Pavón.

## 5. Conclusions

The Spanish expedition to the Viceroyalty of Peru brought to light a great diversity of Peruvian plants. We have compiled 6493 specimens, of which 2327 are useful plants used by native indigenous people and would later serve as medicines in Lima or Spain, including some, such as the *Cinchona* tree or palm trees used in construction, *Phytelephas macrocarpa* or psycholeptics such as *Psychotria tinctoria* Ruiz & Pav. (= *Palicourea tinctoria* Schult.), whose uses and effects are still being studied in the 21st century.

With this work, we have described the plant communities where the RSBEVP studied many species, whether they had uses and which possible ethnic groups used them. For this, we have drawn up maps with the observation localities with their corresponding bioclimatology, observing the type of vegetation in their environment. Most of the plants collected or observed come from the thermo- and mesotropical bioclimatic belts, both from the eastern forests and from the Peruvian coast. In fact, the expedition studied species from all of the habitats of Peru, but those in the eastern rainforests were extensively collected due to the political interest in finding species of the genus *Cinchona*. It was precisely in these forests that they obtained most of the useful plants, so it seems that the knowledge of the Pano, Arawaw and Jíbaro ethnic families was very important.

From a taxonomic point of view, the genera most studied by Ruiz, Pavón and Tafalla include *Cinchona, Erythroxylum, Nolana, Polylepis, Theobroma* and *Valeriana*.

The study of the flora observed during past expeditions can provide a new method for carrying out new ethnobotanical research in areas close to the expedition itineraries and for analyzing and discovering new bioactive molecules.

It may even be possible to use the expedition routes as a tourist attraction and for the conservation of historic forests.

## Figures and Tables

**Figure 1 biology-12-00294-f001:**
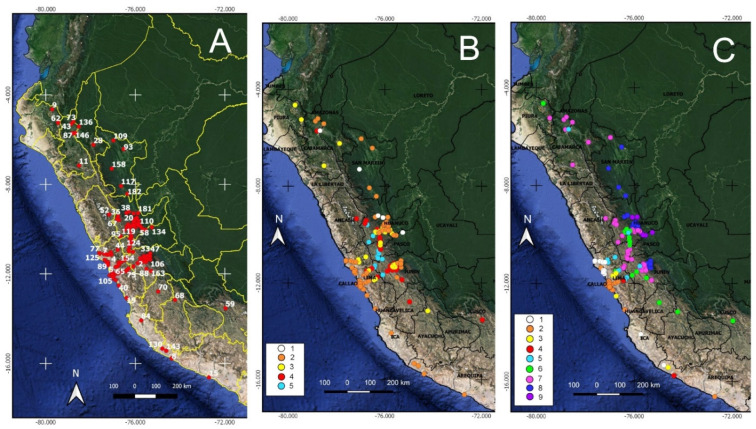
(**A**) Localities of the expedition of Ruiz and Pavón throughout Peru; (**B**) bioclimatic belt in each locality (1. infratropical; 2. thermotropical; 3. mesotropical; 4. supratropical; 5. orotropical); (**C**) ombrotype in each locality (1. ultra−hyperarid; 2. hyperarid; 3. arid; 4. semiarid; 5. dry; 6. subhumid; 7. humid; 8. hyperhumid; 9. ultra−hyperhumid).

**Figure 2 biology-12-00294-f002:**
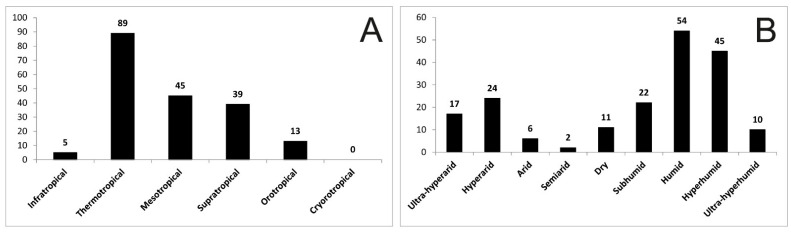
(**A**) Number of localities of the expedition per bioclimatic belt; (**B**) number of localities of the expedition per ombrotype.

**Figure 3 biology-12-00294-f003:**
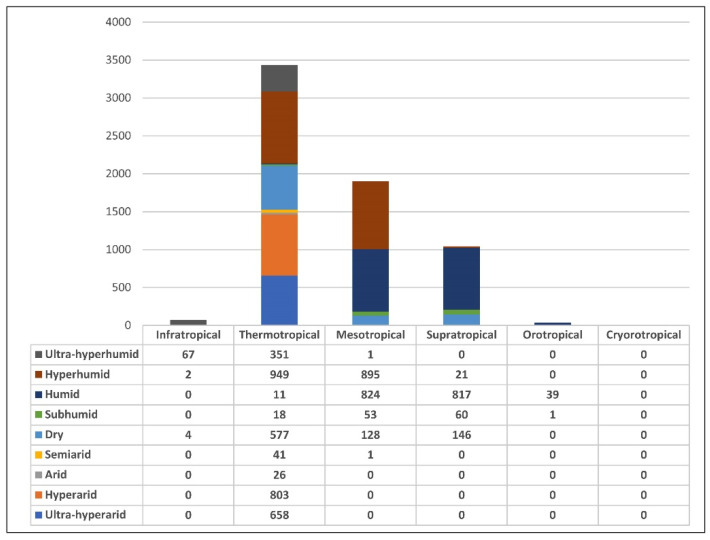
Frequencies of the plants of the RSBEVP expedition among bioclimatic belts and ombrotypes.

**Figure 4 biology-12-00294-f004:**
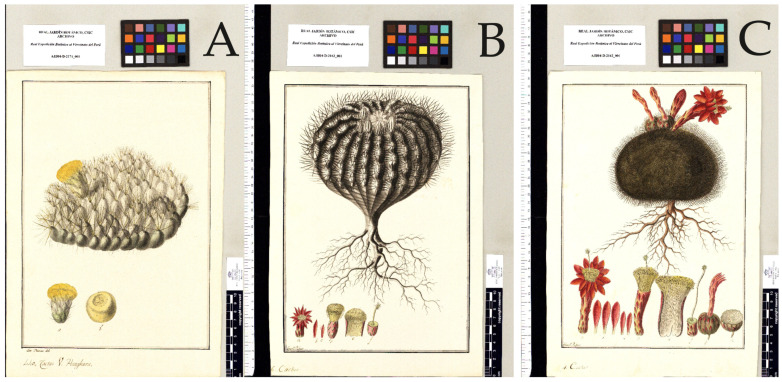
Drawings of Cactaceae from the RSBEVP, which are kept in the Royal Botanical Garden of Madrid (reproduction with permission). (**A**) *Cactus* Ruiz & Pav. (probably *Austrocylindropuntia floccosa* (Salm-Dyck) F. Ritter, (**B**) *Cactus melocactus* L. (= *Oroya peruviana* (K.Schum.) Britton & Rose) and (**C**) *Cactus melocactus* L. (probably *Matucana haynii* (Otto ex Salm-Dyck) Britton & Rose).

**Figure 5 biology-12-00294-f005:**
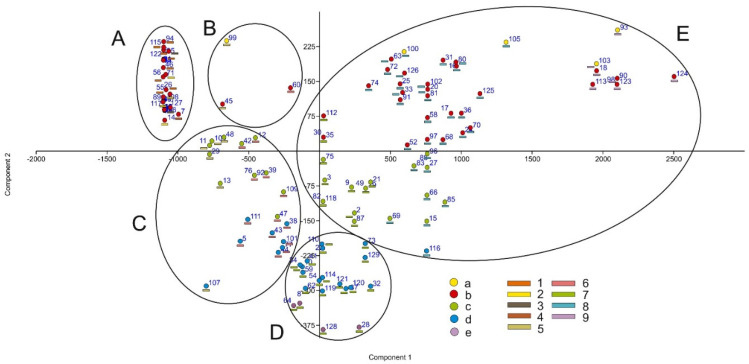
PCA with the localities of useful plants distributed among bioclimatic belts (coloured dots) and ombrotypes (coloured rectangles). Bioclimatic belts: a. infratropical; b. thermotropical; c. mesotropical; d. supratropical; e. orotropical. Ombrotypes: 1. ultra−hyperarid; 2. hyperarid; 3. arid; 4. semiarid; 5. dry; 6. subhumid; 7. humid; 8. hyperhumid; 9. ultra−hyperhumid.

**Figure 6 biology-12-00294-f006:**
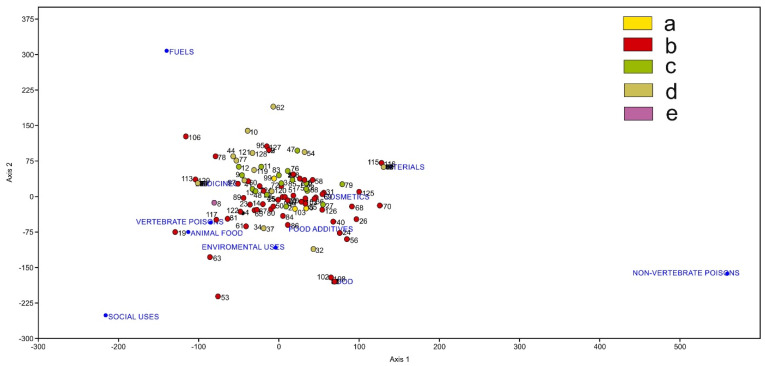
DCA showing the distribution of localities with different frequencies of useful plants with respect to their uses. Bioclimatic belts: a. infratropical; b. thermotropical; c. mesotropical; d. supratropical; e. orotropical.

**Figure 7 biology-12-00294-f007:**
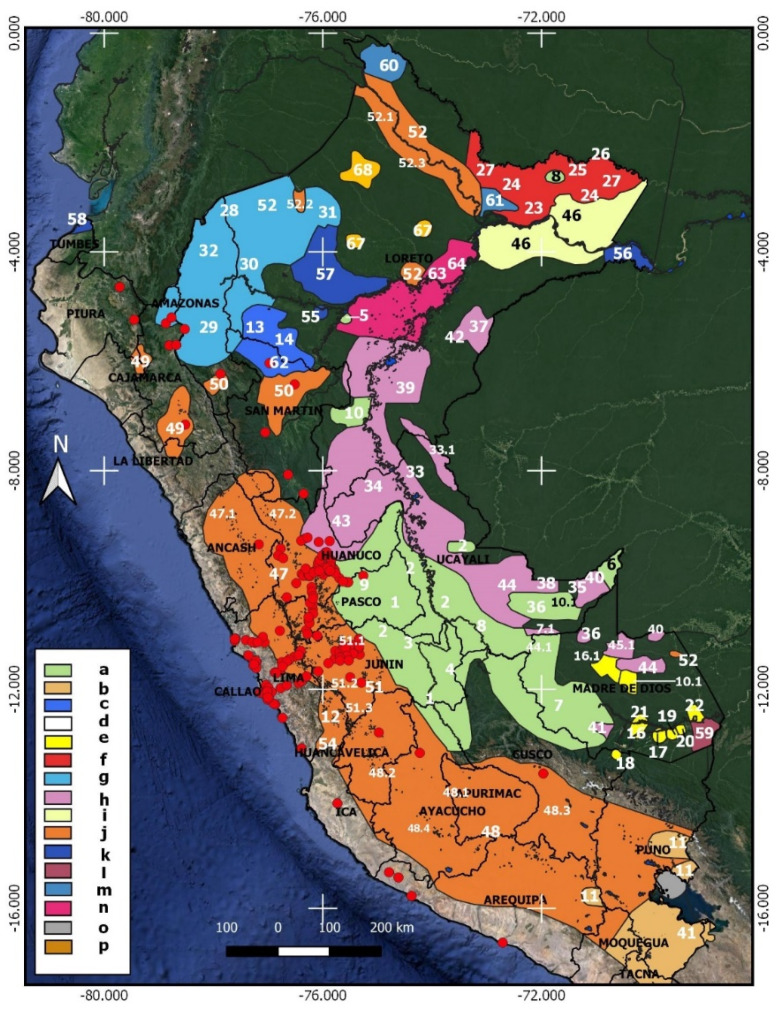
Map showing expedition localities (red dots) and their correspondence with ethnic group families (colored polygons) and ancestral ethnicities (white numbers). Families of ethnic groups: a. Arawaw, b. Aru, c. Cahuapana, d. Castilian, e. Harakmbut, f. Huitoto, g. Jibaro, h. Pano, i. Peba−Yagua, j. Quechua, k. Unclassified, l. Tacana, m. Tucano, n. Tupi−Guaraní, o. Uro−Chipaya, p. Zaparo. Ethnic groups: 1. Asháninka, 2. Ashéninka, 3. Atiri, 4. Caquinte, 5. Chamicuro, 6. Madija, 7. Matsiguenga and Noshaninkajeg (7.1), 8. Resígaro, 9. Yanesha, 10. Yine and Kapexuchi−Nawa (10.1), 11. Aymara, 12. Jakaru, 13. Campo−Piyapi, 14. Shiwlu, 15. Castilian, 16. Amarakaeri, 17. Arasaire, 18. Huachipaeri, 19. Kisamberi, 20. Pukirieri, 21. Sapiteri, 22. Toyoeri, 23. Dyo’ xaiya or Ivo’tsa, 24. Meneca, 25. Miamuna, 26. Muinane, 27. Murui, 28. Achuar, 29. Awajun, 30. Candoshi−Shappra, 31. Jibaro, 32. Shuar−Wampis, 33. Iscobaquebu, 34. Joni, 35. Junikuin, 36. Masrronahua, 37. Matsés, 38. Morunahua and Morunahu (38.1), 39. Nuquencaibo, 40. Onicoin, 41. Parquenahua, 42. Pisabo, 43. Uni and Cashibo−Cacataibo (43.1), 44. Yaminahua, 45. Yora, 46. Yihamwo, 47. Ancash−Yaru, Vicos (47.1) and Yaruvilcas (47.2), 48. Ayacucho−Cusco, Chancas (48.1), Chopccas (48.2), Quero (48.3) and Wari (48.4), 49. Cañaris−Cajamarca, Cajamarca (49.1) and Cañaris (49.2), 50. Chachapoyas−Lamas and Llacuash (50.1), 51. Jauja−Huanca, Huancas (51.1), Tarumas (51.2) and Xauxas (51.3), 52. Napo−Pastaza−Tigre, Alamas (52.1), Ingas (52.2) and Quichua (52.3), 53. Santarrosino and Kichwaruna (53.1), 54. Yauyos, 55. Aguano, 56. Duüxügu, 57. Kachá Edze, 58. Walingos, 59. Ese’ejja, 60. Aido pa, 61. Maijuna, 62. Monichis, 63. Cocama−Cocamilla, 64. Omagua, 65. Uro, 66. Iquito, 67. Ite’chi, 68. Tapueyocuaca.

**Figure 8 biology-12-00294-f008:**
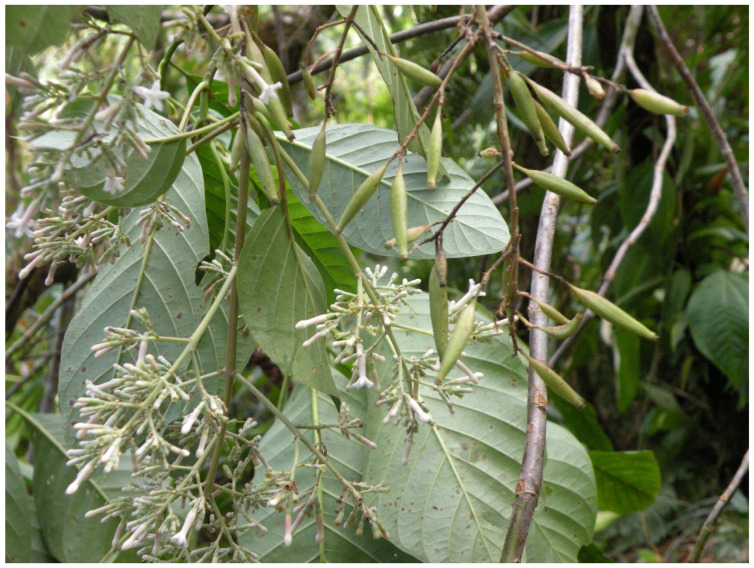
*Cinchona micrantha* Ruiz & Pav. growing in a thermotropical ultra−hyperhumid Amazonian forest from southern Peru (San Gabán, Puno. Photograph by E.L.P.).

**Table 1 biology-12-00294-t001:** Synthesis of Peruvian vegetation linked to bioclimatic belts and ombrotypes, based on Galán de Mera et al. [35,36]. Species names according to IPNI [29].

Bioclimatic Belt	Ombrotype	Vegetation Type	Floristic Combination	Geographic Emplacement
Infratropical	Semiarid	Xerophytic vegetation with columnar cacti	*Bursera graveolens* *Espostoa lanata* *Loxopterygium huasango* *Neoraimondia arequipensis* *Parkinsonia praecox*	Occidental slopes of northern Peru
Infratropical	Semiarid	Xerophytic vegetation with columnar cacti	*Armatocereus rauhii* subsp. *balsasensis* *Browningia altissima* *Parkinsonia praecox*	Oriental slopes of northern Peru
Infratropical	Humid–ultra-hyperhumid	Rainforests	*Euterpe precatoria* *Iriartea deltoidea* *Mauritia flexuosa*	Oriental slopes of southern Peru and Amazonian plain
Thermotropical	Dry	Xerophytic forests	*Croton ruizianus* *Vachellia macracantha*	Occidental slopes of northern Peru
Thermotropical	Subhumid	Mesophytic forests	*Annona cherimola* *Escallonia pendula* *Vachellia macracantha*	Occidental slopes of northern Peru
Thermotropical	Dry	Xerophytic forests	*Cedrela kuelapensis* *Diplopterys leiocarpa* *Eriotheca ruizii* *Vachellia macracantha*	Oriental slopes of northern Peru
Thermotropical	Hyperarid	Desert ephemeral vegetation	*Nolana gayana* *Palaua rhombifolia* *Tetragonia crystallina*	Central coastal desert
Thermotropical	Arid	Xerophytic vegetation with columnar cacti	*Neoraimondia arequipensis* *Orthopterygium huaucui* *Weberbauerocereus winterianus*	Occidental slopes of Central Peru
Thermotropical	Hyperarid	Desert ephemeral vegetation	*Hoffmannseggia miranda* *Nolana scaposa* *Palaua weberbaueri*	Southern coastal desert
Thermotropical	Arid	Xerophytic vegetation with columnar cacti	*Browningia candelaris* *Weberbauerocereus weberbaueri*	Occidental slopes of southern Peru
Thermotropical	Ultra-hyperhumid	Rainforests	*Cinchona micrantha* *Cyathea subincisa* *Iriartella setigera*	Oriental slopes of southern Peru
Mesotropical	Subhumid–humid	Mesophytic forests	*Alnus acuminata* *Oreopanax eriocephalus* *Vallea stipularis*	Northern Peru
Mesotropical	Semiarid	Xerophytic vegetation with columnar cacti	*Armatocereus matucanensis* *Espostoa melanostele*	Occidental slopes of Central Peru
Mesotropical	Dry	Mesophytic shrubs	*Barnadesia blakeana* *Ophryosporus peruvianus*	Occidental slopes of Central Peru
Mesotropical	Arid-semiarid	Xerophytic vegetation with columnar cacti	*Balbisia verticillata* *Corryocactus brevistylus* *Weberbauerocereus weberbaueri*	Occidental slopes of southern Peru
Mesotropical	Hyperhumid	Rainforests	*Fuchsia sanctae-rosae* *Oreopanax eriocephalus* *Smallanthus parviceps*	Oriental slopes of southern Peru
Supratropical	Subhumid–humid	Mesophytic forests	*Barnadesia dombeyana* *Buddleja incana* *Polylepis racemosa*	Northern Peru
Supratropical	Dry–subhumid	Mesophytic shrubs and forests	*Aristeguietia ballii* *Aristeguietia discolor* *Polylepis racemosa*	Central Peru
Supratropical	Semiarid–dry	Xerophytic shrubs and forests	*Diplostephium tacorense* *Fabiana stephanii* *Parastrephia quadrangularis* *Polylepis rugulosa*	Southern Peru
Supratropical	Humid	Rainforests	*Berberis peruviana* *Mutisia cochabambensis* *Polylepis incarum*	Oriental slopes of southern Peru
Orotropical	Humid–hyperhumid	Grasslands	*Ascidiogyne sanchezvegae* *Festuca huamachucensis* *Puya fastuosa*	Northern Peru
Orotropical	Dry–subhumid	Grasslands	*Festuca chrysophylla* *Festuca dolichophylla* *Parastrephia lucida*	Southern Peru
Orotropical	Dry–subhumid	Grasslands and peatlands	*Calamagrostis rigida* *Festuca dolichophylla* *Distichia muscoides*	Central Peru
Cryorotropical	Subhumid–humid	Vegetation due to gelifraction processes	*Stellaria cryptantha* *Werneria ciliolata*	Central Peru
Cryorotropical	Dry–subhumid	Vegetation due to gelifraction processes	*Mniodes coarctata* *Nototriche obcuneata* *Werneria poposa*	Southern Peru

**Table 2 biology-12-00294-t002:** Correspondence between bioclimatic belts, vegetation and phytochemistry of some species of the expedition.

Bioclimatic Belt	Ruiz and Pavón’s Name (Updated Name According to IPNI [28])	Vegetation Type	Phytochemistry	Reference
Infratropical Thermotropical	*Cinchona micrantha*	Rainforests	Quinoline alkaloids	[60]
Thermotropical	*Narcissus amancaes* (*Ismene amancaes*)	Desert ephemeral vegetation	Galanthamine-type alkaloids	[61]
Thermotropical Mesotropical	*Cinchona lutea* *Cinchona obovata* *Cinchona pallescens* *Cinchona purpurea* (*Cinchona pubescens*)	Rainforests	Quinoline alkaloids	[60]
Thermotropical Mesotropical	*Erythroxylum coca*	Rainforests	Tropane alkaloids	[45]
Mesotropical, Supratropical	*Monnina polystachya*	Mesophytic forests	Steroids	[62]
Mesotropical Supratropical	*Periphragmos uniflorus* (*Cantua buxifolia*)	Mesophytic shrubs and forests	Steroids, phenolic compounds, flavonoids	[63]
Mesotropical Supratropical	*Mutisia acuminata*	Mesophytic shrubs and forests	Flavonoids	[64]
Mesotropical Supratropical	*Polylepis racemosa*	Mesophytic shrubs and forests	Triterpenoids	[65]
Orotropical	*Valeriana lanceolata*	Grasslands	Iridoids	[66]
Orotropical	*Valeriana pilosa*	Grasslands	Iridoids	[67]

**Table 3 biology-12-00294-t003:** Correspondence between some texts from the diaries, current localities and ethnic groups.

Diary	Chapter	Pages	Text	Current Locality	Ethnic Group
B	VI	56	* **Los Indios** * *de esta Provincia [Huarochirí] se dedican a la Arriería y laboreo de Minas*	Huarochirí (Lima)	Jakaru (12)
B	VII	69–71	*confinan con las Montañas de los **Indios infieles** [Santa Rosa de Ocopa]*	Mountains near Santa Rosa de Ocopa (Junín)	Jauja-Huanca (51)
B	VIII	80	* **Indios bárbaros** * *como era de presumir, por no distar aquel paraje mas que seis leguas de Chanchamayo donde en la actualidad se hallaban congregados*	La Merced (Junín)	Ashéninka (2)
B	VIII	91–92	*se une con el Marañon por las tierras de los **Yndios Omaguas** [no contacto]*	Iquitos (Loreto)	Omagua (64)
B	IX	98	*este corto Pueblo de **Indios [Acomayo]***	Acomayo (Huánuco)	Ancash-Yaru (47)
B	X	101–102	*Los Cascarilleros o Recolectores de Quina se apropiaron los Ranchos así que * * **los Indios se trasladaron a Pueblo Nuevo** *	Pueblo Nuevo (Huánuco)	Uni and Cashibo−Cacataibo (43)
B	X	102	* **Los Indios de los Lamas,** * *gastan para subir hasta Cuchero dos o tres meses*	Lamas (San Martín)	Chachapoyas−Lamas (50)
B	XVI	139	* **aquellos indios [Pozuzo]** *	Pozuzo (Pasco)	Yanesha (9)
J	XII	75	*por el Este con las Montañas de **los Indios barbaros** [respecto a Provincia de Tarma]*	Tarma (Junín)	Jauja−Huanca (51)
J	XII	77	*el territorio de los **Indios Lamas** [sobre el origen del Marañón]*	Lamas (San Martín)	Chachapoyas−Lamas (50)
J	XII	81	* **catequizar á aquellos Indios,** * *se ha fundado en este año de 1779 una Población con su fortaleza en el sitio de Chanchamayo,*	La Merced (Junín)	Ashéninka (2)
J	XXI	140	*Panao; dista del Valle ocho leguas, tiene sobre **200 vecinos Indios***	Panao (Huánuco)	Yanesha (9)
J	XLVII	294	*Vivirían gustosos como salvages y andarían desnudos como los demás **Indios barbaros de su «Nación Carapacha»** ó desnuda*	Huamalíes (Huánuco)	Uni and Cashibo−Cacataibo (43)
J	LV	353	* **Los Indios de los Pueblos de Acomayo, Panao** * *y principalmente los de **Pillao***	San Pablo de Pillao (Huánuco)	Uni and Cashibo−Cacataibo (43)

B: diary by Barreiro; J: diary by Jaramillo. Words in bold and highlighted in gray signify the place in the text where native people appear. The numbers in brackets match with those in Figure 7.

## Data Availability

Not applicable.

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
