# Peer review of "Biogeographical Relationships and Diversity in the Peruvian Flora Reported by Hipólito Ruiz and José Pavón: Vegetation, Uses and Anthropology"

_biology, 2023, doi:10.3390/biology12020294_

Round 1

Reviewer 1 Report

The article is about the expeditions of Ruiz and Pavon and the authors try to find the biogeographical relationships and diversity in the Peruvian flora through these data sources. It is very interesting for me to find the localities of previous studies and to analyze the data provided by milestone studies in this field. I have made some notes on the file as comments. I think the paper is relevant and could be accepted in its present form. in fact, I am not aware of the Peruvian flora so if the identification needs more consideration I am not familiar with it.

Author Response

Dear Reviewer,

Thank you very much for your kind and stimulating reply. We have taken into consideration all your comments noted in the margin of the text.

Yours sincerely,

The authors

Reviewer 2 Report

The authors have used various statistical methods, such as the multivariate statistical technique used to find the main factors or gradients in large species-rich, sparsely distributed data matrices to identify and manage ecological data to overcome distortions in data analyses.

Line 96. In carrying out the study of Peruvian place names did reliance on names assigned from 19thcentury typonymies create any limitations? Do the QGIS typonymies incorporate the information collected by the Yale Peruvian Expeditions of 1911, 1912, and 1914-1915, under the auspices of the National Geographic Society? Were the Expeditions records reviewed by the authors?

Line 464. What is meant by “promising plants”? Promising to whom - for what purpose(s)? How dothe perception of what are "promising plants" in the 18th century contrast with that of the 21st century?

The conclusion would benefit from suggestions for future research projects, given the foundation the author’s study has laid down. What can we learn from this study for disciplines currently exploring remaining uncharted areas, even oceanic realms?

Author Response

Dear Reviewer,

Thank you very much for your comments, which are really constructive and aimed at improving the manuscript. In red you have our response to your suggestions.

Reviewer:

The authors have used various statistical methods, such as the multivariate statistical technique used to find the main factors or gradients in large species-rich, sparsely distributed data matrices to identify and manage ecological data to overcome distortions in data analyses.

Line 96. In carrying out the study of Peruvian place names did reliance on names assigned from 19thcentury typonymies create any limitations? Do the QGIS typonymies incorporate the information collected by the Yale Peruvian Expeditions of 1911, 1912, and 1914-1915, under the auspices of the National Geographic Society? Were the Expeditions records reviewed by the authors?

Authors: All the toponymies were revised by constructing the expedition's itineraries with the references given in supplement 3. The itineraries could be made by compiling all the localities in the expedition's diaries. Many of them coincide with the current Peruvian toponymy, so we were able to register their coordinates in QGIS. All the localities of the itineraries are shown in the maps in figure 1.

Except for some particular cases that have required further study, such as Pozuzo (present-day Tilingo), most of the place names from the time of the Expedition (18th-19th centuries) remain unchanged in their diction and location. There are cases of old "populations" or small border fortifications that have now disappeared, but thanks to the profuse description given in the diaries, we have been able to locate them, as is the case of San Teodoro de Colla.

The Yale and National Geographic Society expeditions, led by Hiram Bingham, were focused on the area around the Urubamba River valley, Cuzco, and evidently Machu Picchu. These territories were never explored by the members of the Botanical Expedition to the Viceroyalty of Peru, only the use of certain plants in Cuzco is referred to. Bingham reached Arequipa, where the Boyden Observatory of Harvard University was located (until 1927), and continued parallel to the coast towards Coropuna, without reaching Camaná or Lomas de Pongo and the Acarí River (explored by Tafalla).

Reference: Bingham, H. & Brownson, H. F. (1922). Inca land: explorations in the highlands of Peru. Houghton Mifflin.

It is worth mentioning that the Yale team mapped the Peruvian coast and other territories in a general way, mentioning only a few main cities, or in some maps simply the towns as "town".

Reference: Bowman, I. (1916). The Andes of southern Peru: geographical reconnaissance along the seventy-third meridian.

Reviewer: Line 464. What is meant by “promising plants”? Promising to whom - for what purpose(s)? How dothe perception of what are "promising plants" in the 18th century contrast with that of the 21st century?

Authors: We changed 'promising plants' to 'useful plants', which is more understandable and apparently more widespread.

We have answered your questions by lengthening the sentence in the text and saying that some plants have still being studied and used in the same way as they were in Ruiz & Pavón's time.

Reviewer: The conclusion would benefit from suggestions for future research projects, given the foundation the author’s study has laid down. What can we learn from this study for disciplines currently exploring remaining uncharted areas, even oceanic realms?

Authors: We have added in the conclusions what may be the interest of this type of study.

Sincerely, The authors

Reviewer 3 Report

The work of Ruiz and Pavón is an important legacy for the flora of Peru, and this study provides an original and valuable account of their monumental expedition. The manuscript is a pleasure to read, the concept is interesting, the arguments clear and the methods sound. The colonial past of herbarium collections is currently a relevant topic, and it is very positive to see this study conducted by Spanish and Peruvian research groups. Such joint initiatives are particularly welcome and can cover the kind of gaps discussed by the authors at the end of the introduction.

I have a few suggestions to help improve the manuscript.

I wonder whether the specimens, drawings (beautiful examples given in Figure 4) and the diaries are digitized and publicly available online. How can the interested reader look them up? Providing some (perma)links leading to the original material of the expedition would be useful.

The authors can consider providing at the beginning of the manuscript brief biographic information about Hipólito Ruiz and José Pavón. For example, who were these two men, who sent them to Peru, perhaps what was their personal motivation for this journey, if details are known. This could be done by enriching the paragraph in lines 43-48. This framework information will help international readers to better connect with the history of the expedition and the herbarium.

At the end of the introduction (lines 75-85), the authors list some research gaps around this expedition. It is great that with this manuscript they place the missing pieces, demonstrating the value of such international collaborations. I recommend that (part of) this information is transformed into a clear aim of this specific study. The aim is now not obvious in the main body of the text.

The term “plants” is mentioned throughout the manuscript, but this is quite ambiguous. It is not clear whether the authors refer to the number of collected plant species, the number of species they correspond to, or to the original plant names used by the expeditioners. These numbers can be different if the expeditioners collected duplicates of the same species, and possibly under different names. In the conclusions (line 464), it seems that the authors refer to plant specimens, but this should be clarified early on in the manuscript. In the methods, the authors state that they have updated the original plant names given by Ruiz and Pavón. Was this a nomenclatural update? But then in Supplement 1 the plant names are given as “Taxa by Ruiz & Pavón” which implies that these are the original names used by the expeditioners. It would be useful to make clear in the manuscript which are the original names and what was the update done by the authors. And also how many botanical species the 6493 specimens correspond to.

Also, several of the species names in Supplement 1 seem to be indigenous names, that possibly (?) the expeditioners adopted from the locals. Historical names of useful plants are scarce and at the same time very valuable for ethnobotanical studies. If the indigenous plant names recorded by the expeditioners have not been published elsewhere, I wonder if it would be possible to indicate (in Supplement 1?) these names and which ethnic group used each name. I understand that this may go beyond the research goals of this paper, but it would be worthwhile to consider.

Please check Supplement 1 for potential typo’s.

The term “promising plants”, mentioned in several parts of the manuscript, also needs clarification. Based on which criteria did the authors conclude that 2327 “plants” out of 6493 collected specimens are promising, and what are they promising for?

The end of the manuscript is somewhat “dry” and abrupt. The authors could consider adding one or two lines over the wider implications of this research. For example, they end the introduction laying out the knowledge gaps regarding the expedition. They can consider returning to these gaps and sum up in the conclusions how their study has filled them.

Author Response

Dear Reviewer,

Thank you very much for your comments. We have taken most of them into consideration. Please find below a response to your suggestions:

Reviewer:

The work of Ruiz and Pavón is an important legacy for the flora of Peru, and this study provides an original and valuable account of their monumental expedition. The manuscript is a pleasure to read, the concept is interesting, the arguments clear and the methods sound. The colonial past of herbarium collections is currently a relevant topic, and it is very positive to see this study conducted by Spanish and Peruvian research groups. Such joint initiatives are particularly welcome and can cover the kind of gaps discussed by the authors at the end of the introduction.

I have a few suggestions to help improve the manuscript.

I wonder whether the specimens, drawings (beautiful examples given in Figure 4) and the diaries are digitized and publicly available online. How can the interested reader look them up? Providing some (perma)links leading to the original material of the expedition would be useful.

Authors: Most of the herbarium material and drawings from the expedition are digitized in the Global Plants by JSTOR portal. This was added in the text.

Reviewer: The authors can consider providing at the beginning of the manuscript brief biographic information about Hipólito Ruiz and José Pavón. For example, who were these two men, who sent them to Peru, perhaps what was their personal motivation for this journey, if details are known. This could be done by enriching the paragraph in lines 43-48. This framework information will help international readers to better connect with the history of the expedition and the herbarium.

Authors: Added this information.

Reviewer: At the end of the introduction (lines 75-85), the authors list some research gaps around this expedition. It is great that with this manuscript they place the missing pieces, demonstrating the value of such international collaborations. I recommend that (part of) this information is transformed into a clear aim of this specific study. The aim is now not obvious in the main body of the text.

Authors: The final paragraph with the aims of the work has been changed.

Reviewer: The term “plants” is mentioned throughout the manuscript, but this is quite ambiguous. It is not clear whether the authors refer to the number of collected plant species, the number of species they correspond to, or to the original plant names used by the expeditioners. These numbers can be different if the expeditioners collected duplicates of the same species, and possibly under different names. In the conclusions (line 464), it seems that the authors refer to plant specimens, but this should be clarified early on in the manuscript. In the methods, the authors state that they have updated the original plant names given by Ruiz and Pavón. Was this a nomenclatural update? But then in Supplement 1 the plant names are given as “Taxa by Ruiz & Pavón” which implies that these are the original names used by the expeditioners. It would be useful to make clear in the manuscript which are the original names and what was the update done by the authors. And also how many botanical species the 6493 specimens correspond to.

Authors: The authors are referring to both collected and observed plants appearing in the diaries and other materials, as indicated in the chapter on material and methods. The names used in Supplement 1 correspond to the original names of Ruiz & Pavon (Taxa by Ruiz & Pavon), although we have added the updated names according to IPNI and the labels of the herbarium sheets of the expedition in the species indicated in the text. Although there are many species where this is not possible.

Reviewer: Also, several of the species names in Supplement 1 seem to be indigenous names, that possibly (?) the expeditioners adopted from the locals. Historical names of useful plants are scarce and at the same time very valuable for ethnobotanical studies. If the indigenous plant names recorded by the expeditioners have not been published elsewhere, I wonder if it would be possible to indicate (in Supplement 1?) these names and which ethnic group used each name. I understand that this may go beyond the research goals of this paper, but it would be worthwhile to consider.

Authors: In Supplement 1 there are no indigenous names, they are scientific names from Ruiz & Pavón. The historical names of the uses of the plants are quite inaccurate because in many cases the illness to which they have being applied is not really known, so they are not very useful for ethnobotanical studies either. Moreover, our work is not an ethnobotanical study. In the same way, it is very difficult to indicate the ethnic group to which the expeditionaries attribute the use of a plant, since they almost always used very general names, such as 'Indian infidels' (see Table 3), but not the name of an ethnic group. Precisely the aim of our work is to indicate the possible ethnic groups from which the plants came, as well as the type of vegetation from which they were obtained, since nothing was known about this.

Reviewer: Please check Supplement 1 for potential typo’s.

Authors: We have rechecked all the Supplements.

Reviewer: The term “promising plants”, mentioned in several parts of the manuscript, also needs clarification. Based on which criteria did the authors conclude that 2327 “plants” out of 6493 collected specimens are promising, and what are they promising for?

Authors: We changed 'promising plants' to 'useful plants', which is more understandable and apparently more widespread. The authors' criteria for choosing useful plants are given in Material & Methods: “In addition, the use assigned to the plants by the indigenous people of the localities through which the expedition passed was noted and translated according to Cook [29].”

Reviewer: The end of the manuscript is somewhat “dry” and abrupt. The authors could consider adding one or two lines over the wider implications of this research. For example, they end the introduction laying out the knowledge gaps regarding the expedition. They can consider returning to these gaps and sum up in the conclusions how their study has filled them.

Authors: We have changed the 'Conclusions' paragraph.

Responses to comments on the pdf document:

Lines 43-46: Reviewer: Long sentence consider splitting.

Authors: The sentence has been shortened.

Line 98: Reviewer: "the list of plants [...] which were updated”: what kind of update do the authors refer to? Is it nomenclatural update according to presently accepted botanical species names or something else?

Authors: We have added that they were updated nomenclaturally.

Line 144: Reviewer: What is meant with "localities with promising species", please explain

Authors: We have changed this concept to 'useful plants'.

Lines 398-402: Reviewer: long and complex sentence consider splitting.

Authors: We have split this sentence.

Line 424: Reviewer: what is meant with: "with their altitudinal succession"?

Authors: We have changed 'succession' to 'distribution'.

Lines 426-427: Reviewer: this information regarding the purpose of the expedition is missing from the introduction, please considering moving it there.

Authors: We have not removed this sentence from here, but we have added this topic in the introduction.

Line 429: Reviewer: unclear grammar, consider rephrasing

Authors: Yes, of course. Changed the sentence.

Lines 432-435: Reviewer: the authors mix too many things in this sentence and the meaning gets lost. Consider splitting and rephrasing

Authors: The sentence has been rewritten and split.

Line 436: Reviewer: the term "active principles" is somewhat ambiguous, maybe a more suitable term would be "bioactive compounds"

Authors: We have changed 'active principles' to 'bioactive compounds'

Sincerely, The authors
